# Optimisation and Characterisation of Novel Angiotensin-Converting Enzyme Inhibitory Peptides Prepared by Double Enzymatic Hydrolysis from *Agaricus bisporus* Scraps

**DOI:** 10.3390/foods11030394

**Published:** 2022-01-29

**Authors:** Rui Wang, Jianmin Yun, Shujuan Wu, Yang Bi, Fengyun Zhao

**Affiliations:** College of Food Science and Engineering, Gansu Agricultural University, Lanzhou 730070, China; 15002561189@163.com (R.W.); 1665565391@163.com (S.W.); mm18198017645@163.com (Y.B.); zfy15214014980@163.com (F.Z.)

**Keywords:** angiotensin-converting enzyme, inhibitory peptides, double-enzyme hydrolysis, binding mechanism, peptide characterisation, mushroom peptides

## Abstract

Food-derived hypotensive peptides have attracted attention in the field of active peptide research in recent years. In this study, based on ACE inhibition rate and using the Box–Behnken central combination design principle to optimise the process of ACE inhibitor peptides prepared by double-enzyme hydrolysis. The amino acid sequences of ACE inhibitor peptides were determined by liquid chromatography mass spectrometry (LC-MS/MS), and their binding to ACE was studied by molecular docking. The optimal processing conditions were 1:1 alkaline protease: compound protease, pH was 8.43, enzymolysis temperature was 44.32 °C, and enzymolysis time was 3.52 h. Under these conditions, the ACE inhibition rate reached 65.12%, and the inhibition rate after separation and purification was 80.68% (IC_50_ = 0.9 mg/mL). Three novel peptides with ACE inhibitory activity were detected by LC-MS/MS, with sequences LVYP (Leu-Val-Tyr-Pro), VYPW(Val-Tyr-Pro-Trp) and YPWT(Tyr-Pro-Trp-Thr). Molecular docking revealed that the three novel peptides all established hydrogen bonds with the S1(Tyr523, Glu384, Ala354) and S2 (His353) pockets of ACE. Among them, LVYP, VYPW and YPWT, respectively, formed eleven hydrogen bonds, six hydrogen bonds and nine hydrogen bonds with ACE. The study revealed that these peptides have the potential for the development of novel ACE inhibitor drugs and provide a new avenue for high-value utilisation of mushrooms scraps.

## 1. Introduction

*Agaricus**bisporus* is an edible basidiomycete mushroom that is widely cultivated worldwide, with production accounting for 54% of total edible fungi [1]. It has a high protein content, accounting for 3.5~4.5% of the fresh weight, making it a potential protein resource [2]. In addition to fresh sales, mushrooms are processed into canned products. During processing, about 10% of edible mushroom stems, defective products and scraps are discarded, which not only increases environmental pollution but also represents a waste of resources [3]. These scraps are high in protein and low in fat and have the potential for the preparation of bioactive peptides.

Biologically active peptides are beneficial for life activities or have specific physiological activity [4]. Angiotensin-converting enzyme (ACE) is a key rate-limiting enzyme in the regulation of human blood pressure. ACE inhibitory peptides can prevent ACE from catalysing the hydrolysis of bradykinin, thereby facilitating normal vasoconstriction [5]. They can also inhibit the conversion of angiotensin I into angiotensin II, which has the ability to constrict blood vessels, thereby lowering blood pressure [6]. In addition, food-derived ACE inhibitory peptides are different from synthetic antihypertensive drugs because they are safe, largely free from side-effects, and suitable for long-term use. Therefore, food-derived ACE inhibitory peptides have been attracting attention in the field of active peptide research in recent years [7].

At present, the main methods for obtaining ACE inhibitory peptides from edible fungi include direct extraction methods such as water extraction, ammonium sulphate precipitation and methanol precipitation [8]. The main edible fungi species are *Tricholoma giganteum* [9], *Hypsizygus marmoreus* [10], *Agrocybe aegirit* [11,12], *Ganoderma Lucidum Karst* [13] and *Pleurotus ostreatus* [14], and many more cultivated edible fungi fruiting bodies are also consumed. However, direct extraction of ACE inhibitory peptides is limited due to the low active peptide content of edible fungi fruiting bodies, and different types of edible fungi can differ significantly [15]. Therefore, the potential for obtaining novel ACE inhibitory peptides is relatively low. The use of enzymatic hydrolysis to prepare ACE inhibitory peptides can not only make fuller use of protein resources in raw materials but also greatly increase the possibility of obtaining new ACE inhibitory peptides [16]. However, few studies have been reported on the preparation of ACE inhibitory peptides from *A. bisporus* by dual-enzyme digestion.

Therefore, in the present work, we used *A. bisporus* as raw material and selected ACE inhibition rate and proteolysis as indicators. After screening proteases, the process conditions were optimised for preparing ACE inhibitor peptides from mushroom protein by double-enzyme hydrolysis. Macroporous resin and gel chromatography were employed to separate and purify the obtained crude peptide solution, liquid chromatography tandem mass spectrometry (LC-MS/MS) was used to determine the amino acid sequences of the ACE inhibitory peptides, and their stability was investigated. Finally, molecular docking was performed to explore the ACE inhibitory mechanism of active peptides. The results provide a scientific basis and theoretical support for the development of potential blood pressure-lowering functional foods and drugs and provide a new avenue for the high-value utilisation of mushrooms scraps.

## 2. Materials and Methods

### 2.1. Experimental Materials

Samples of *Agaricus bisporus* mixed scraps were collected from Tianshui Zhong Xing Fungi Technology Co., Ltd., (Tianshui, China).

### 2.2. Extraction of Protein from A. bisporus Scraps

Extraction of protein was performed as described previously [17] and assessed using the following formula:(1)$$\mathrm{Extraction}\;\mathrm{rate}\;\mathrm{of}\;\mathrm{protein}\;\left(\%\right)=\frac{pro_{c}}{pro_{s}}\times 100\%$$
where *pro_c_* is the extracted protein content and *pro_s_* is the protein content in mushroom scrap samples.

### 2.3. Determination of ACE Inhibition Rate 

The ACE inhibition rate was determined as described previously [18] with slight modifications. The substrate solution was prepared by dissolving 1.0 mmol·L^−1^ FAPGG(N-[3-(2-furylacryloyl)]-L-phenylalanyl-glycyl-glycine) in Tris-HCl (50 mmol·L^−1^) (purchased from Sigma Aldrich) containing 0.3 mol·L^−1^ NaCl at pH 7.5 and placed in a 37 °C water bath. A 10 μL sample of enzyme solution was placed in a 96-well plate, 150 μL of substrate solution was added, the plate was immediately placed in an enzyme calibrator, and the absorbance at 340 nm was measured every 30 s for 30 min. A 10 μL volume of buffer solution was used instead of enzyme solution as a blank control, and 10 μL of 0.25 U·mL^−1^ ACE solution was used instead of enzyme solution as a negative control. The change in absorbance value (ΔA) was plotted against time and the slope was calculated using the following formula:(2)$$\mathrm{Inhibition}\;\mathrm{rate}\;\mathrm{of}\;\mathrm{ACE}\;\left(\%\right)=\left(1-\frac{{\Delta A}_{Samples}}{{\Delta A}_{Blank\;control}}\right)\times 100\%$$

The peptide concentration inhibiting 50% ACE activity (defend as IC_50_) was calculated by plotting the ACE inhibition percentage against different peptide concentrations. Experiments were performed in triplicate, and IC_50_ values of purified peptides were compared with that reported for captopril (used as a positive control).

### 2.4. Determination of the Degree of Protein Hydrolysis

Referring to the method of Adler–Nissen [19] with a slight modification, 0.125 mL of enzymatic hydrolysis solution and 1 mL of 0.1% (*w*/*v*) TNBS (purchased from Sigma Aldrich) was added to a small tube containing 1 mL of phosphate buffer (the pH value was 8.2 ± 0.02), placed in a 50 °C constant temperature water bath in the dark, shaken for 60 min, and cooled at room temperature for 20 min. Then, a 2 mL sample of 0.1 mol/L HCl was added to stop the reaction, and measure the absorbance at 340 nm. The degree of protein hydrolysis was calculated using the following equation:(3)$$DH=\frac{{\mathrm{AN}}_{2}-{\mathrm{AN}}_{1}}{\mathrm{Npb}}\times 100\%$$
where AN_1_ is the content of amino nitrogen before protein hydrolysis, AN_2_ is the content of amino nitrogen after protein hydrolysis, and Npb is the protein content of the protein substrate. The values of AN_1_ and AN_2_ can be calculated from the absorption curve of the standard sample at 340 nm (L-leucine is generally used as the standard sample).

### 2.5. Protease Screening

Powdered mushroom protein was weighed at a substrate concentration of 5% by mass fraction and enzymatically digested using alkaline protease, complex protease, flavour protease, papain, and neutral protease (purchased from Shanghai Yuanye Bio-Technology Co., Ltd., Shanghai, China). According to the degree of protein hydrolysis and the ACE inhibition rate, two proteases with superior preparation effects were selected for compounding.

### 2.6. Single Factor Experimental Design

#### 2.6.1. Addition Ratio of Double-Enzyme Hydrolysis

The substrate concentration of mushroom protein was formulated to 5%. The two enzymes selected were added respectively at the ratios of 2:1, 1:1, 1:2, 1:3, 1:4 and 1:5 for alkaline protease: complex protease, the addition amount of the two enzymes was 6000 u·g^−1^. The enzymatic digestion time was 2 h, the enzymatic digestion temperature was 55 °C, and the pH of enzymatic digestion was 7.5. The ACE inhibition rate and protein hydrolysis degree were calculated according to Equations (2) and (3), respectively.

#### 2.6.2. Determination of the Optimal Temperature for Enzymatic Digestion

According to the optimal ratio of the dual enzymes and their optimal addition amounts determined in Section 2.6.1, the mushroom protein substrate concentration was 5%, the enzymatic digestion duration was 2 h, the enzymatic pH was 7.5, and the enzymatic digestion temperature was tested at 40, 45, 50, 55 and 60 °C. The ACE inhibition rate and degree of protein hydrolysis were calculated according to Equations (2) and (3), respectively.

#### 2.6.3. Determination of the Optimal pH for Enzymatic Hydrolysis

According to the optimal protease addition ratio determined in Section 2.6.1 and the optimal enzymatic hydrolysis temperature determined in Section 2.6.2, the mushroom protein substrate concentration was 5%, the enzymatic hydrolysis duration was 2 h, and the pH of enzymatic hydrolysis was tested at 6.5, 7.0, 7.5, 8.0 and 8.5. The ACE inhibition rate and protein hydrolysis degree were calculated according to Equations (2) and (3), respectively.

#### 2.6.4. Determination of the Optimal Enzymatic Digestion Duration

According to the optimal addition ratio of protease determined in Section 2.6.1 and the optimal enzymatic digestion temperature and pH determined in Section 2.6.2 and Section 2.6.3, the mushroom protein substrate concentration was 5%, and the enzymatic digestion duration was tested at 0.5, 1.5, 2.5, 3.5, 4.5 and 5.5 h. The ACE inhibition rate and degree of proteolysis were calculated according to Formulas (2) and (3), respectively.

### 2.7. Response Surface Experimental Design

According to the results of the above single factor experiments, taking the ACE inhibition rate as the response value and using the Box–Behnken central combination experiment design as the principle, a response surface analysis method with three factors and three levels was designed to optimise the process for preparing ACE inhibitors. The setting of factors and levels are shown in Table 1.

### 2.8. Separation and Purification

#### 2.8.1. Purification of Mushroom ACE Inhibitor Peptides Using Macroporous Resin (Purchased from Shanghai Yuanye Bio-Technology Co., Ltd., Shanghai, China)

##### Screening of the Macroporous Resin

Four different macroporous adsorption resins (DA201-C, XAD1600, XAD7HP and AB-8, purchased from Shanghai Yuanye Bio-Technology Co., Ltd., Shanghai, China) were tested in static adsorption experiments, and the optimal resin was identified according to the adsorption rate calculated using the following formula: (4)$$\mathrm{Adsorption}\;\mathrm{rate}\;\left(\%\right)=\frac{A-B\;}{A}\times 100\%$$
where A is the protein concentration of stock solution, and B is the protein concentration of hydrolysate after adsorption.

##### Dynamic Adsorption and Desorption Experiments

Dynamic adsorption and desorption experiments were performed as described previously [20] with slight modifications. The chromatographic column (purchased from Shanghai Yuanye Bio-Technology Co., Ltd., Shanghai, China) was φ2.5 × 60 cm in size with a 200 mL DA201-C resin bed volume, a loading volume of 200 mL, a loading flow rate of 1.6 mL/min, a deionised water flushing flow rate of 2 mL/min, an ethanol (purchased from Shanghai Yuanye Bio-Technology Co., Ltd., Shanghai, China) elution flow rate of 2 mL/min, and 5 min/tube collection. After loading the sample at a defined flow rate, the column was washed with deionised water to remove unabsorbed enzymatic hydrolysate. The absorbance of each test tube was measured at a wavelength of 280 nm. When the conductivity of the wash fractions was almost the same as that of the water or no longer changed, elution was performed with ethanol and eluate fractions were collected at 5 min/tube. Elution peak fractions were combined, concentrated to remove ethanol, and the desorption rate was calculated using the following formula:(5)$$\mathrm{Desorption}\;\mathrm{rate}\left(\%\right)=\frac{\alpha }{\beta \;-\;\gamma }\times 100\%$$
where α is the peptide content of the desorption solution, β is the peptide content of the original solution, and γ is the peptide content of the wash solution.

#### 2.8.2. Gel Chromatography Separation

Sephadex G-100 (purchased from Shanghai Yuanye Bio-Technology Co., Ltd., Shanghai, China) was used to fractionate the purified ACE inhibitory peptides, and distilled water was used as the eluent for elution. Eluted fractions were collected in separate tubes (3 mL/tube) and collected based on peaks. The absorbance of fractions at 280 nm (A_280nm_) was measured using an ultraviolet spectrophotometer.

### 2.9. Determination of Molecular Mass by HPLC 

For the determination of molecular mass by HPLC (Shimadzu LC-20A, Japan), a TSK(TOSOH) Gel 2000 SWXL 300 × 7.8 mm column was employed, with a mobile phase of 30:70:0.1 (*v*/*v*/*v*) acetonitrile:water:trifluoroacetic acid, a detection wavelength of 280 nm, a flow rate of 1 mL/min, and a column temperature of 30 °C. Bovine serum albumin (Mw67000), VB12 (Mw 1335) and oxidised glutathione (Mw 614) served as mixed standards. A standard curve was plotted according to the above chromatographic conditions (Figure 1).

As shown in Figure 1, the elution durations corresponding to the appearance of elution peaks were 1.885, 6.234 and 7.225 min. Values were plotted against elution time, and the linear regression equation obtained using the least squares method was y = −0.3793x + 5.4946 (where x represents the peak time and y represents the logarithm of the molecular mass), and the regression coefficient R^2^ = 0.9987.

### 2.10. Sequence Determination

Sequence determination of ACE inhibitor peptides was performed by Beijing Biotech Pack Biotechnology Co., Ltd., (www.biotech-pack.com (accessed on 25 April 2021). Beijing, China).

### 2.11. Molecular Docking

The 3D structure of the screening target protein was retrieved and downloaded from the PDB database, and the protein was processed by dewatering and adding hydrogens. The Define Receptor function in the Tools Explorer module set the processed protein molecule as the receptor and used the CDocker tool to perform molecular docking of the ligand and receptor.

### 2.12. Stability Verification

#### 2.12.1. pH Stability of ACE Inhibitory Peptides

A series of buffers at pH 2.0 to 10.0 were prepared (pH 3.5~5.5 = 0.1 mol/L sodium acetate buffer; pH 6.0~7.0 = 0.1 mol/L phosphate buffer; pH 8.0~10.0 = 0.1 mol/L carbonate buffer). A 2 mL sample of 0.5 mg/mL ACE inhibitory peptides were placed in buffers at different pH values, incubated at 4 °C for 12 h, and the method described in Section 2.3 was used to measure ACE inhibitory activity. Relative activity was calculated by comparing with the original measured value, and the stability of ACE inhibitory peptides at different pH values was assessed. 

#### 2.12.2. Temperature Stability of ACE Inhibitory Peptides

ACE inhibitory peptides were preincubated at different temperatures (25~95 °C) for 30 min, and ACE inhibitory activity was tested using the method described in Section 2.3 after rapid cooling. The results were compared with the original measured values to obtain relative activity, and the stability of the ACE inhibitory peptides at different temperatures was evaluated.

#### 2.12.3. In Vitro Digestion of ACE Inhibitory Peptides

ACE inhibitory peptides were digested as described previously [21] with some modifications. A 5 mg sample of pepsin/trypsin in 0.1 mol/L KCl-HCl buffer pH 2.0/0.1 mol/L phosphate buffer pH 8.0 was prepared, and the volume was adjusted to 100 mL to prepare 0.05 mg/mL of pepsin/trypsin solution. Lyophilized powder from separated peaks (peptide content) was weighed and dissolved in 5 mL of 0.05 mg/mL pepsin/trypsin solution (E/S = 1:200) and shaken in a water bath at 37 °C for 3~4 h. The reaction was terminated by heating in a water bath at 100 °C for 5 min to inactivate the enzyme. The pH of the reaction solution was adjusted to 8.3 (no pH adjustment for trypsin), the supernatant was centrifuged at 10,000× *g* for 10 min, and the supernatant was used to determine ACE inhibition activity.

A 2 mL sample of the pepsin-digested solution was mixed with 2 mL of 0.05 mg/mL trypsin solution and shaken at 37 °C for 4 h (E/S = 1:200). Subsequently, the reaction was terminated by heating in a water bath at 100 °C for 5 min to inactivate the enzyme and centrifuged at 10,000× *g* for 10 min. The supernatant was used to determine ACE inhibitory activity.

## 3. Results

### 3.1. Screening of Proteases

In order to obtain suitable proteases for hydrolysing *A. bisporus* proteins, five proteases (compound protease, papain, flavour protease, alkaline protease and neutral protease) were subjected to enzymatic hydrolysis tests, and the results are shown in Figure 2.

All five proteases resulted in ACE inhibitory activity following enzymatic hydrolysis of mushroom protein. Compared with the direct extraction method, the degree of protein hydrolysis and the ACE inhibition rate were significantly altered after hydrolysis with each of the five protein hydrolases. The ACE inhibition rate was lowest after enzymolysis by neutral protease and flavour protease and highest with alkaline protease and complex protease, and the degree of protein hydrolysis followed the same pattern. Therefore, alkaline protease and complex protease were selected for hydrolysis of mushrooms scraps to prepare ACE inhibitory peptides.

### 3.2. Single Factor Testing

#### 3.2.1. Effects of Compound Ratio on ACE Inhibition Rate

The ACE inhibition rate and degree of proteolysis first increased, then decreased, with decreasing compound ratio (Figure 3). When the alkaline protease:composite protease ratio was 1:1, the ACE inhibition rate reached its maximum (63.20%). However, the degree of proteolysis was low (only 41.66%) and increased with decreasing compound ratio up to its maximum (54.37%) at a compound ratio of 1:4. Compared with the direct extraction method (24.36%), the ACE inhibition rate under different compound ratios was generally improved. This may be because hydrolysis yielded new active peptides, which significantly increased the ACE inhibition rate. The two-enzyme method did indeed promote the production of novel active peptides. Therefore, an alkaline protease:compound protease ratio of 1:1 was selected for subsequent experiments.

#### 3.2.2. Effects of pH on ACE Inhibition Rate

From Figure 4, it can be seen that both the ACE inhibition rate and the degree of protein hydrolysis first increased then decreased with increasing pH. At a pH of 8.5, both parameters peaked (ACE inhibition rate = 64.89%, degree of protein hydrolysis = 42.36%). Therefore, a pH of 8.5 was selected as the central value for subsequent response surface experiments.

#### 3.2.3. Effects of Enzymatic Digestion Temperature on ACE Inhibition Rate

Both the ACE inhibition rate and degree of protein hydrolysis first increased then decreased with increasing enzymatic digestion temperature (Figure 5). Both parameters peaked at an enzymatic digestion temperature of 45 °C (ACE inhibition rate = 64.40%, degree of protein hydrolysis = 45.33%). Therefore, 45 °C was selected as the central value for subsequent response surface experiments.

#### 3.2.4. Effects of Enzymatic Digestion Duration on ACE Inhibition Rate

From Figure 6, it can be seen that both the degree of protein hydrolysis and ACE inhibition rate continued to increase up to an enzymatic digestion duration of 3.5 h. The ACE inhibition rate of the hydrolysate began to decrease beyond 3.5 h, but the degree of protein hydrolysis continued to rise, then stabilised. Therefore, an enzymatic digestion duration of 3.5 h was chosen as the central value for subsequent response surface experiments. After 3.5 h, the degree of hydrolysis and ACE inhibition rate were 30.23% and 65.39%, respectively.

### 3.3. Response Surface Optimisation

Based on the single factor test results, the main factors affecting the preparation of ACE inhibitory peptides from mushrooms were pH, enzymatic digestion duration and enzymatic digestion temperature. Therefore, these three influencing factors were selected to carry out response surface experiments. The response surface results are shown in Table 2, and analysis of variance is shown in Table 3.

According to the response surface test data, a quadratic polynomial equation was derived as follows:Y = +69.38 − 2.79 * A − 2.76 * B + 1.08 * C − 0.28 * A * B + 0.88 * A * C + 2.51 * B * C − 9.73 * A^2^ − 9.55 * B^2^ − 7.35 * C^2^,
where Y is the ACE inhibition rate (%), A is the pH of the enzymatic digestion solution, B is the enzymatic digestion temperature, and C is the enzymatic digestion duration. It can be seen from Table 3 that the model was extremely significant (F = 8.89, *p* < 0.0001). The coefficient of determination of the model (R^2^) was 0.9977, and the lack of fit term was not significant, indicating that the model fitted well and was relatively reliable. The pH, enzymatic digestion temperature and enzymatic digestion duration were all significant, and the order of significance was pH > enzymatic digestion temperature > enzymatic digestion duration. Thus, pH had the greatest influence on the ACE inhibition rate of mushroom scraps protein, followed by enzymatic digestion temperature, and enzymatic digestion duration has the least influence. Interactions between enzymatic digestion duration and enzymatic digestion temperature, and between enzymatic digestion duration and pH, were significant (*p* < 0.05; Figure 7).

Figure 7 shows how pH, temperature, and duration interact to influence the ACE inhibition rate. The steepness of the upper spatial surface of the network graph reflects the degree of influence of the independent variable on the response value of the ACE inhibition rate, and the elliptical eccentricity of the contour map at the lower part of the image reflects the effect of interactions of the factors. The vertices of each curved surface represent the optimal conditions for the factors affecting the enzymolysis. The theoretical optimal enzymolysis conditions predicted by Design Expert 8.0 software were pH 8.43, enzymatic digestion temperature 44.32 °C, and enzymatic digestion duration 3.52 h, giving an ACE inhibition rate of 65.12%. Under these conditions, the verification test was carried out, and the measured ACE inhibition rate was 64.97%, similar to the predicted value, indicating that the test was reliable.

### 3.4. Separation and Purification

#### 3.4.1. Purification of ACE Inhibitory Peptides Using Macroporous Resin

##### Screening of Macroporous Resin

From Figure 8, it can be seen that DA201-C macroporous resin was superior to other resins in terms of adsorption speed and adsorption efficiency. Thus, DA201-C was selected as the purification medium for subsequent purification experiments.

##### Dynamic Adsorption and Washing of Enzymatic Hydrolysates

The calculated desorption rate with 75% ethanol was 77.86%, and that for 25% ethanol was 69.38%. Thus, 75% ethanol had a higher desorption rate and could reduce the loss of enzyme hydrolysate, and hence, 75% ethanol was chosen for desorption. When the elution volume was between 0 and 100 mL, the absorbance value and activity of peptides were lower, indicating adsorption of the peptide by the resin (Figure 9). After this point, the absorbance value began to rise rapidly, indicating that the resin had reached saturation. When the eluent volume was 300~800 mL, the activity of inhibitory peptides was highest, and the next purification step was explored.

#### 3.4.2. Separation of ACE Inhibitory Peptides by Gel Chromatography

Purified peptides were further separated by SephadexG-100 (Figure 10). It was found that the activity first increased then decreased as the purification progressed. The highest ACE inhibitory peptide activity was observed in the fourth and fifth tubes, with activity values of 78.79% and 76.79%, respectively, and the extraction rate was 6.678% (calculated as mushroom protein). HPLC showed that the active fraction was composed of three peptides with molecular masses of 490.279, 563.274 and 565.254 Da.

### 3.5. Comparison of the Activity of ACE Inhibitory Peptides before and after Purification and Separation

The IC_50_ value is the concentration of ACE inhibitor that decreases activity by 50%, and the lower the value, the higher the activity of the active peptide [22]. It can be seen from Figure 11 that as the separation and purification proceeded, the activity of ACE inhibitory peptides derived from mushroom gradually increased; the ACE inhibition rate of the enzymatic hydrolysis solution (EHS) was 65.12%; the ACE inhibition rate of the EHS after purification by macroporous resin DA201-C (MR DA201-C) was 72.12%; the activity of ACE inhibitory peptides after isolation by Sephadex G-100 (S G-100) was 80.68%. The final IC_50_ value was 0.9 mg/mL, which is higher than captopril (IC_50_ = 2.2 ng/mL), a drug prescribed as an ACE inhibitor [23]. However, this is lower than the ACE inhibitory activity of peptides derived from *Triticum aestivum* L. (IC_50_ was 2.79 mg/mL) [24] and *Navodonseptentrionalis* (IC_50_ was 1.96 mg/mL) [25]. Therefore, mushroom-derived ACE inhibitory peptides possess great potential for the development of antihypertensive monomer drugs.

### 3.6. Sequence Determination of ACE Inhibitory Peptides

In order to determine the sequences of active peptides, LC-MS/MS was employed (Figure 12). Based on the MS data, peptide sequences were determined through de novo analysis, and the sequences were LVYP, VYPW and YPWT (Figure 13). We searched the database of biologically active peptides (http://bis.zju.edu.cn/biopepdbr/index.php (accessed on 29 April 2021), http://pepbank.mgh.harvard.edu (accessed on 29 April 2021), http://www.uwm.edu.l/biochemia/ (accessed on 29 April 2021)), and found that the three identified peptides were not previously identified as ACE inhibitors, which confirmed that the three tetrapeptides were novel ACE inhibitory peptides.

### 3.7. Molecular Docking

The sequences of the three ACE inhibitory peptides were LVYP, VYPW and YPWT. Molecular docking showed that both the central six-membered ring and five-membered ring of the overlapping structure of ACE inhibitory peptides engage in Pi-Pi conjugation with His410 of ACE. The specific interaction mode with ACE is shown in Figure 14.

In addition, hydrogen bonds also play an important role in the interaction between inhibitors and ACE. The number and length of hydrogen bonds affect the binding force of inhibitors (Figure 14, left). All three peptides form multiple hydrogen bonds with amino acid residues of ACE.

LVYP, respectively, forms two hydrogen bonds with Ala356 and Asp 358, and one hydrogen bond with Glu403, Arg522, Tyr523, Trp357, Glu384, Ala354 and His387, totalling eleven hydrogen bonds. The molecular docking score is 136 points. VYPW, respectively, forms three hydrogen bonds with His353, and one hydrogen bond with Pro407, Tyr523 and Arg522, totaling six hydrogen bonds. The molecular docking score is 169 points. YPWT, respectively, forms two hydrogen bonds with Ala356 and Tyr360, one hydrogen bond with Ala354, Glu384, Asp358, Tyr523 and Glu403, totalling nine hydrogen bonds. The molecular docking score is 160 points.

The active site of ACE is mainly composed of three active pockets; S1 (Ala354, Glu384 and Tyr523), S2 (Gln281, His353, His513, Lys511 and Tyr520) and S1’ (Glu162) [26]. Some of the amino acid residues that form hydrogen bonds with the tetrapeptides belong to these active pockets. For example, Glu384, Tyr523 and Ala354 belong to the S1 active pockets, and His353 belongs to the S2 active pocket.

Moreover, LVYP and VYPW also interact with Zn2+, and studies have reported that Zn2+ also combines with three other regions (Glu411, His383, His387) to form a twistable tetrahedral structure, which plays a key role in the stability of the compound [27]. Secondly, in addition to hydrogen bonds and interactions with metal ions, van der Waals interactions and Pi-Pi conjugation also play an indispensable role in the stability of the compound (Figure 14, right).

Among the three new peptides, we found that both LVYP and VYPW have the same sequence VYP. The two peptides YPWT and VYPW also shared the same sequence YPW. Therefore, we performed molecular docking for these two shared tripeptides respectively, and the results are shown in Figure 15:

VYP, respectively, forms one hydrogen bond with Glu123, Arg124, Arg124 and Asn85, and forms a conjugate bond with the six-membered ring of tyrosine, and the molecular docking score is 108 points. YPW, respectively, forms one hydrogen bond with Arg124, Ser517, and Glu123 and, respectively, forms a conjugated bond with the six-membered ring of tyrosine, either of the five-membered ring and six-membered ring of tryptophan, and the five-membered ring of proline. The molecular docking score is 138 points. It shows that the above two tripeptides have certain ACE inhibitory activity.

### 3.8. Stability Verification of ACE Inhibitory Peptides

#### 3.8.1. pH Stability of ACE Inhibitory Peptides

The inhibitory activity of ACE was less affected by pH, and its activity remained above 90% in different pH solutions within the range of 2 to 12 (Figure 16), indicating that ACE inhibitory peptides had good tolerance to acid and basic conditions.

#### 3.8.2. Temperature Stability of ACE Inhibitory Peptides

Very little ACE inhibitory activity was lost at various temperatures, with more than 90% of activity retained after treatment (Figure 17). The results showed that the ACE inhibitory peptides had good temperature tolerance.

#### 3.8.3. In Vitro Digestion of ACE Inhibitory Peptides

The results in Figure 18 show that ACE inhibitory peptides were still highly active after hydrolysis by pepsin and trypsin, which reflects the digestion process of food in the body, indicating that ACE inhibitory peptides from mushrooms possess strong resistance to digestion by gastrointestinal enzymes.

## 4. Discussion

The research and development of food-derived active peptides is attracting considerable attention [28,29,30], especially food-borne ACE inhibitory peptides [31,32]. Compared with synthetic ACE inhibitory peptides, food-borne agents have advantages of high food safety, low toxicity and side effects, and no adverse effects on normal blood pressure. They have received extensive attention from researchers for hypertension control and treatment [33].

Edible fungi are a valuable resource for the preparation of active peptides due to their high protein content [34]. Preparation of ACE inhibitory peptides from edible fungi is mainly based on direct extraction [35], but the natural abundance of active peptides in organisms is extremely low, and hence, sourcing ACE inhibitory peptides by this method is limited. In recent years, some researchers have begun to use enzymatic methods to prepare ACE inhibitory peptides from edible fungi. One group [36] and others have optimised the process for preparing ACE inhibitory peptides by hydrolysing protein from *Pleurotus eryngii*, achieving an ACE inhibitory rate of 67.1%. However, hydrolysis by a single protease is limited by the specificity of the enzyme; hence, the degree of hydrolysis of the target protein was relatively low, and the biological activity of the product was easily affected.

In the present study, the activity of ACE inhibitory peptides derived from *A. bisporus* prepared by single-enzyme and double-enzyme methods was compared. It was found that the inhibition rate of ACE inhibitory peptides obtained by single enzymatic hydrolysis was 46.36%, while the inhibitory rate of ACE inhibitory peptides obtained by double enzymatic hydrolysis was 65.12%, an increase of 18.76%, and the hydrolysis effect was superior, consistent with the results of other related studies [37]. Addition of different types of enzymes means that more peptide bonds in the substrate will inevitably be hydrolysed. This greatly increases the content of small peptides in the target protein mixture, which increases the degree of proteolysis. Thus, employing more proteases is beneficial for improving the utilisation of raw materials and obtaining more active peptides.

The key to preparing ACE inhibitor peptides by enzymatic methods is controlling the protease reaction conditions. Inappropriate temperature, duration and pH can affect enzyme activity [38], leading to underutilisation of raw materials, thereby affecting ACE inhibitory activity. Furthermore, excessive hydrolysis may result in some ACE-inhibiting peptides becoming smaller, shorter peptides, and even free amino acids, which in turn affects their ACE-inhibiting activity. Thus, two indicators (ACE inhibition rate and protein hydrolysis) were used to determine the optimal conditions to ensure that raw materials could be fully utilised while obtaining highly active ACE inhibitory peptides.

Food-derived bioactive peptides can only exert their effects when they are absorbed by the intestine into the blood circulation [39]. However, peptide molecules may be affected by gastrointestinal digestive enzymes and gastric juices during the ingestion process, which may decrease activity, and activity may also be affected by storage and circulation conditions (such as temperature). Therefore, whether biologically active peptides can maintain their activity after being affected by the above conditions is very important [40]. The stability verification results for ACE inhibitory peptides obtained in this study demonstrated good tolerance to temperature, pH and gastrointestinal digestive enzymes, indicating great developmental potential.

When preparing biologically active peptides, the separation and purification of peptides is very important [41,42]. Crude peptides derived from edible fungi are complex in composition, usually containing nonenzymatically hydrolysed proteins, peptides with different molecular masses, small peptides and amino acids, salt ions, and other impurities, any of which may affect the activity of the peptides [43]. In the present study, macroporous resin and gel chromatography were used to separate and purify the crude peptides, and after purification, the ACE inhibition rate was increased by 12.10%, and the activity was significantly improved. This may assist the development of antihypertensive peptide monomer drugs.

Molecular docking is an effective method for predicting the binding mode of ligands to receptors based on calculating the interaction energy and intermolecular forces [44]. In order to explore the binding mechanism of ACE inhibitory peptides, we used Discovery Studio software to dock the inhibitory peptides to the ACE active site. Clarification of the binding mechanism of ACE inhibitory peptides can provide guidance for the design and development of new antihypertensive drugs.

Structure–activity relationship studies have shown that potential ACE-inhibiting peptides typically consist of four to six amino acids [44]. Girgih et al. reported that the hydrophobicity of the C-terminal of the peptide played an important role in enhancing the ACE inhibitory activity. They believed that a hydrophobic Val (V) at the third position of the C-terminal, and two hydrophobic Tyr (Y) at the first and second positions of the C-terminal jointly promote the ACE inhibitory activity of a tetrapeptide WVYY [45]. A study on a microalgae ACE inhibitory peptide showed that the C-terminal amino acid was an aromatic or cyclic amino acid, such as Trp, Tyr, etc., which was beneficial to the increase of ACE inhibitory activity [26]. The three novel peptides obtained in this study have similar sequence structure features as above; that is, either end contains hydrophobic or aromatic amino acids, such as Leu (L), Trp (W), Tyr (Y) and Val (V), thereby enhancing its ACE inhibitory activity. Meanwhile, we also found that among the three peptides, LVYP and VYPW, YPWT and VYPW have the same tripeptide sequence VYP and YPW, respectively, and their sequence composition also has the above structural characteristics. That is, either end contains hydrophobic amino acids, such as Trp, Pro and Val, or contain cyclic amino acids, such as Tyr and Trp. It is speculated that these two tripeptide sequences may be the key sequences to maintain the higher activity of the three novel ACE inhibitory peptides.

## 5. Conclusions

In this study, three novel ACE inhibitory peptides were successfully obtained from scraps of *A. bisporus* using dual-enzyme hydrolysis. Under optimal process parameters, the extraction rate of ACE inhibitory peptides was 6.678% (calculated as mushroom protein). This provides a new path for the high-value utilisation of *A. bisporus* scraps. The average activity of the three novel ACE inhibitory peptides was 80.68%, and the IC50 value was 0.9 mg/mL. The amino acid sequences of three novel ACE inhibitory peptides were LVYP and VYPW and YPWT. The stability verification results showed good tolerance to temperature, pH and gastrointestinal digestive enzymes, indicating good potential for the development of drugs for lowering blood pressure. Molecular docking revealed their binding mechanism, which provides guidance for the development of antihypertensive peptide drugs.

## Figures and Tables

**Figure 1 foods-11-00394-f001:**
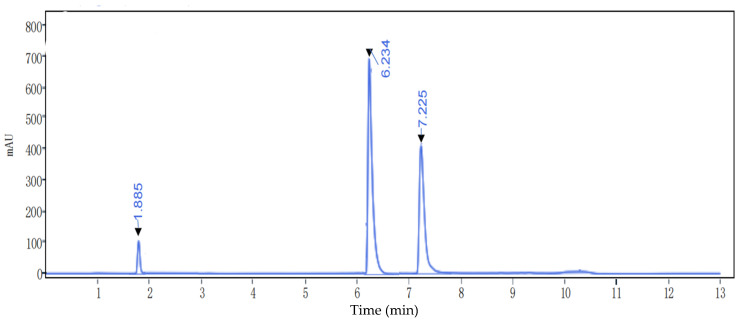
HPLC profiles of three standard samples at 280 nm.

**Figure 2 foods-11-00394-f002:**
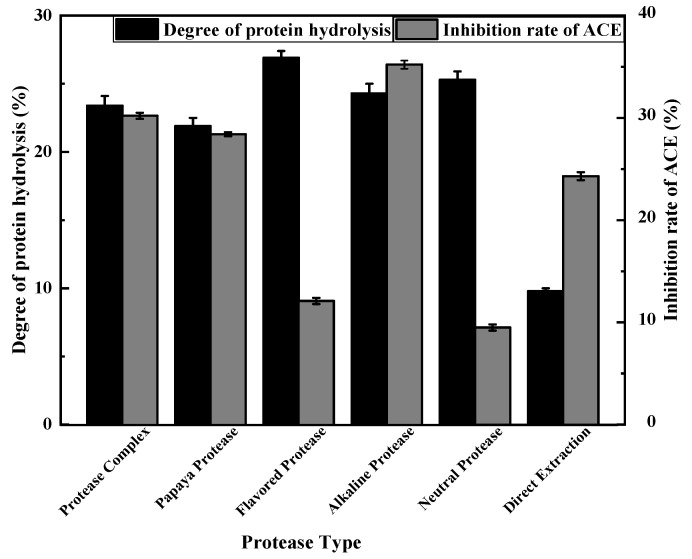
The effect of different proteases on the degree of hydrolysis and the inhibition rate of ACE.

**Figure 3 foods-11-00394-f003:**
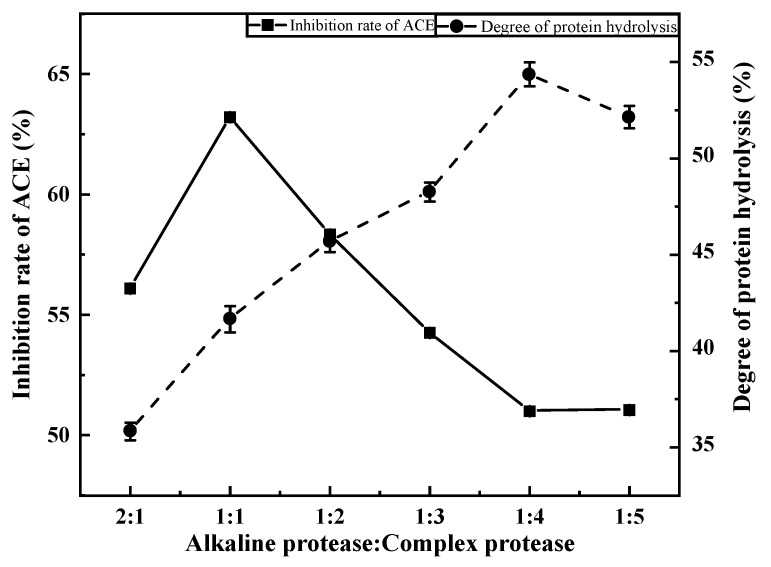
Effect of the addition ratio on the degree of protein hydrolysis and the inhibition rate of ACE.

**Figure 4 foods-11-00394-f004:**
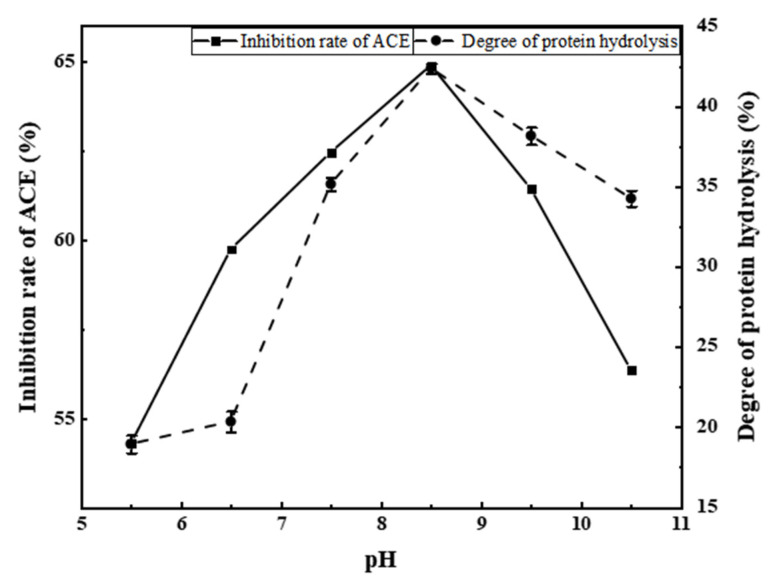
Effect of pH on the inhibition rate of ACE and the degree of protein hydrolysis.

**Figure 5 foods-11-00394-f005:**
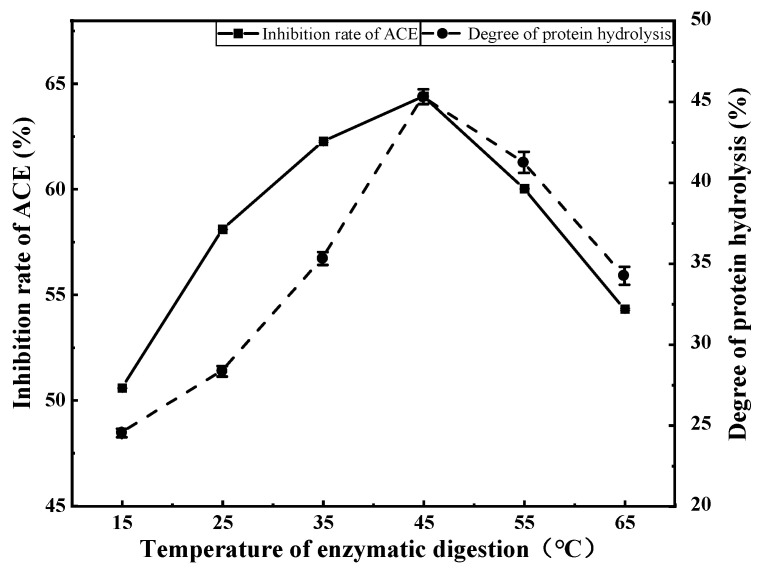
The effect of enzymatic hydrolysis temperature on the inhibition rate of ACE and the degree of protein hydrolysis.

**Figure 6 foods-11-00394-f006:**
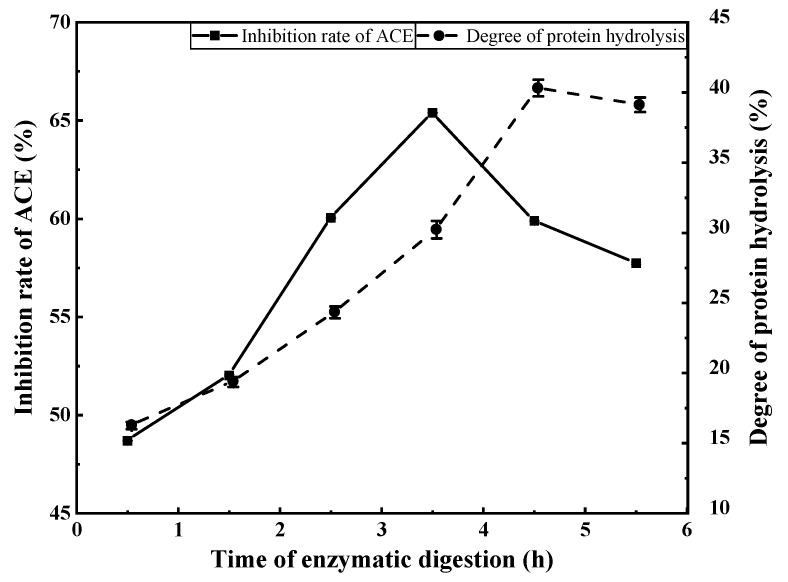
The effect of enzymatic digestion time on the inhibition rate of ACE and the degree of protein hydrolysis.

**Figure 7 foods-11-00394-f007:**
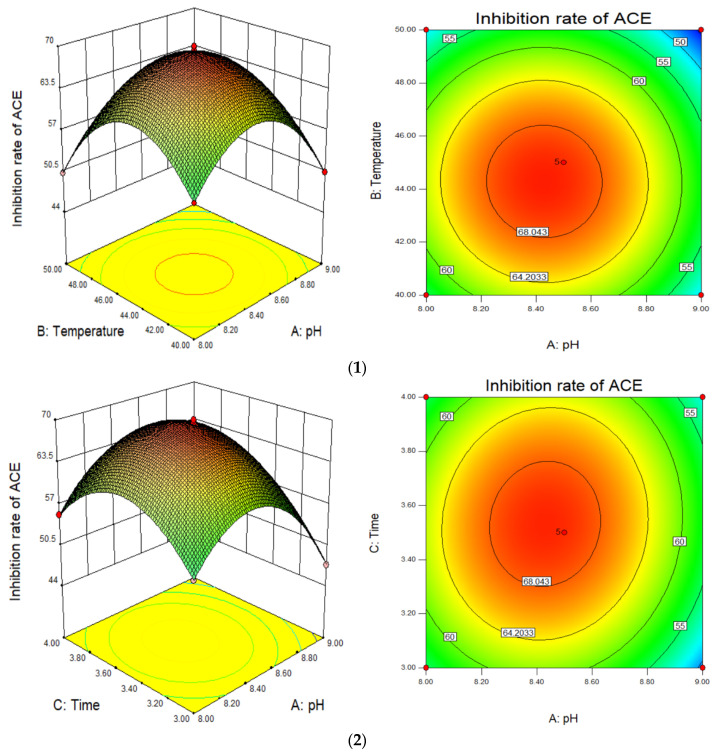
Response surface interaction diagram. (**1**) shows the interaction between pH and temperature. (**2**) shows the interaction between pH and time. (**3**) show the interaction between temperature and time.

**Figure 8 foods-11-00394-f008:**
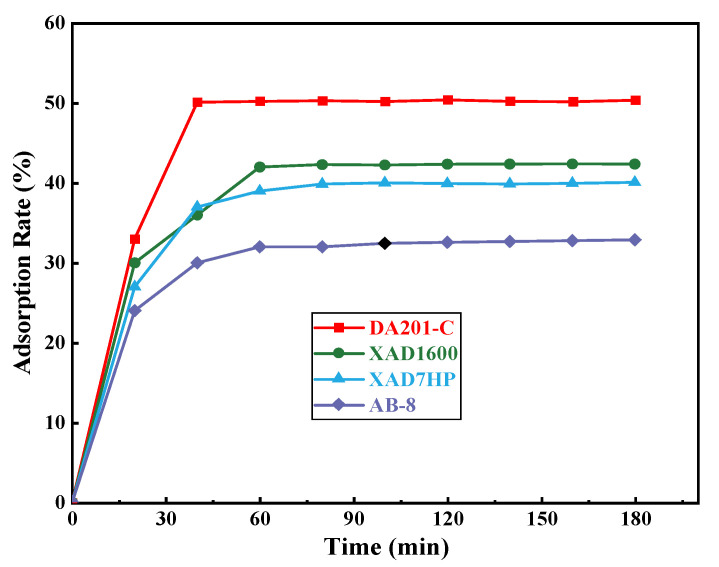
Comparison for purification effects of different macroporous resins. DA201-C, XAD1600, XAD7HP, and AB-8 are different types of macroporous resins, respectively.

**Figure 9 foods-11-00394-f009:**
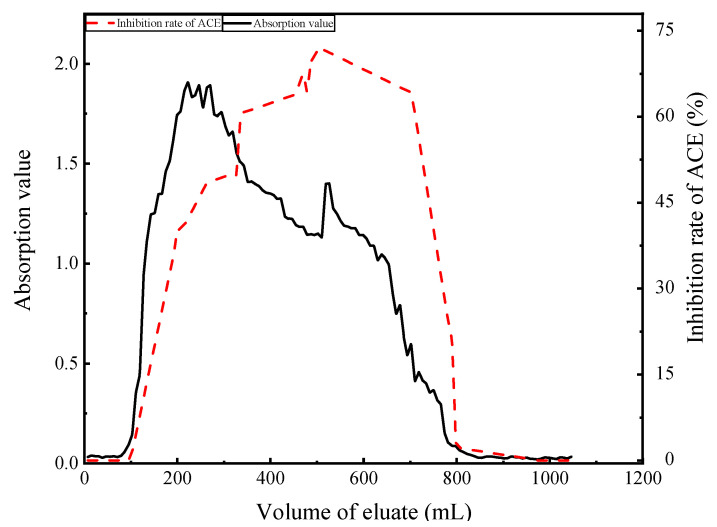
Dynamic adsorption curves of hydrolysates.

**Figure 10 foods-11-00394-f010:**
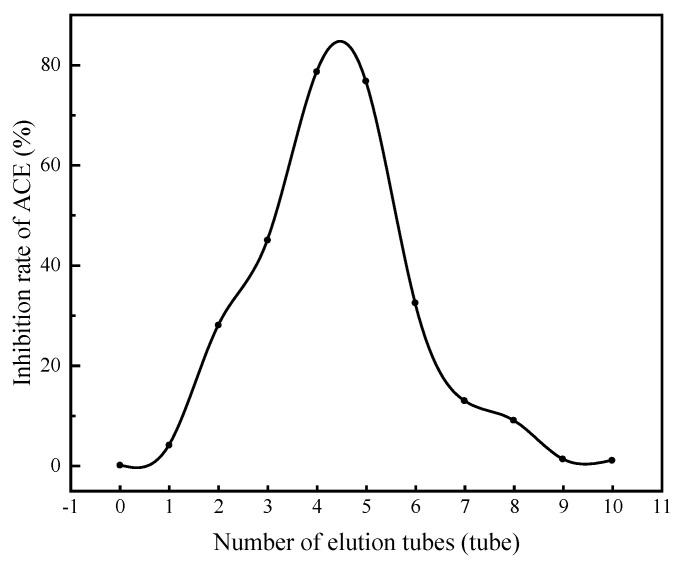
ACE inhibitory activity of different components after separation using SephadexG-100. The purified active peptides were separated by SephadexG-100, and the activity changes in split-peak collection samples were measured, collecting each tube every 5 min.

**Figure 11 foods-11-00394-f011:**
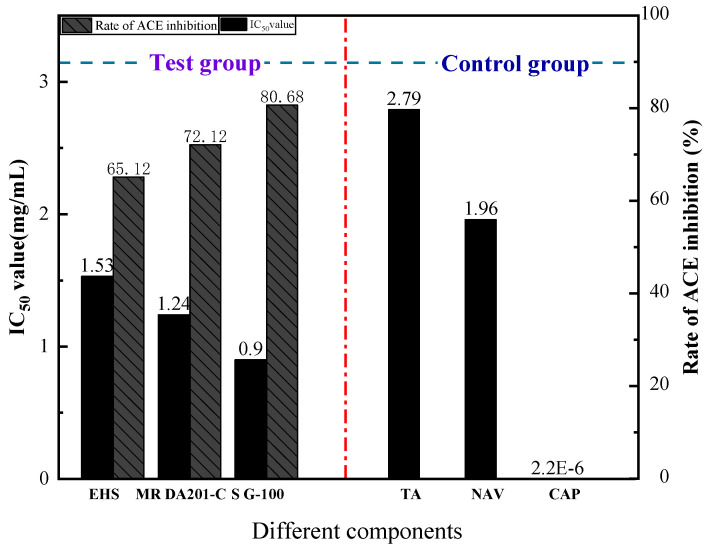
Changes of ACE activity at different preparation stages. “EHS” means enzymatic hydrolysis solution, “MR DA201-C” means the highest active component after purification with macroporous resin DA201-C, and “S G-100” means the highest active component after separation with Sephadex G-100. “TA”, “NAV” and “CAP” stand for *Triticum aestivum, Navodonseptentrionalis* and captopril, respectively.

**Figure 12 foods-11-00394-f012:**
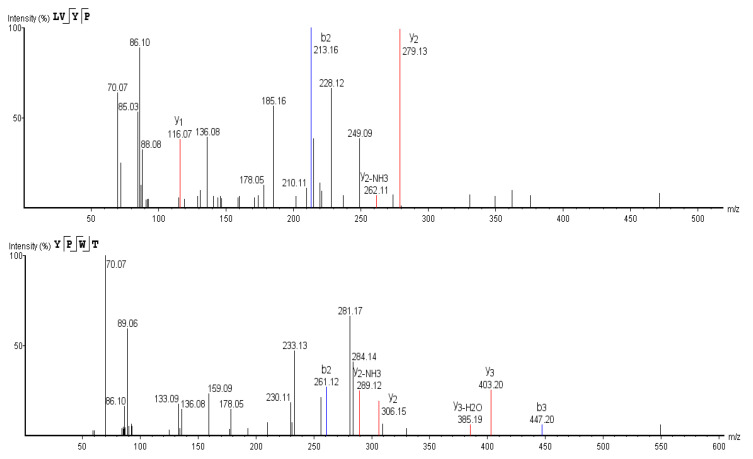
Primary mass spectra of three novel ACE inhibitory peptides.

**Figure 13 foods-11-00394-f013:**
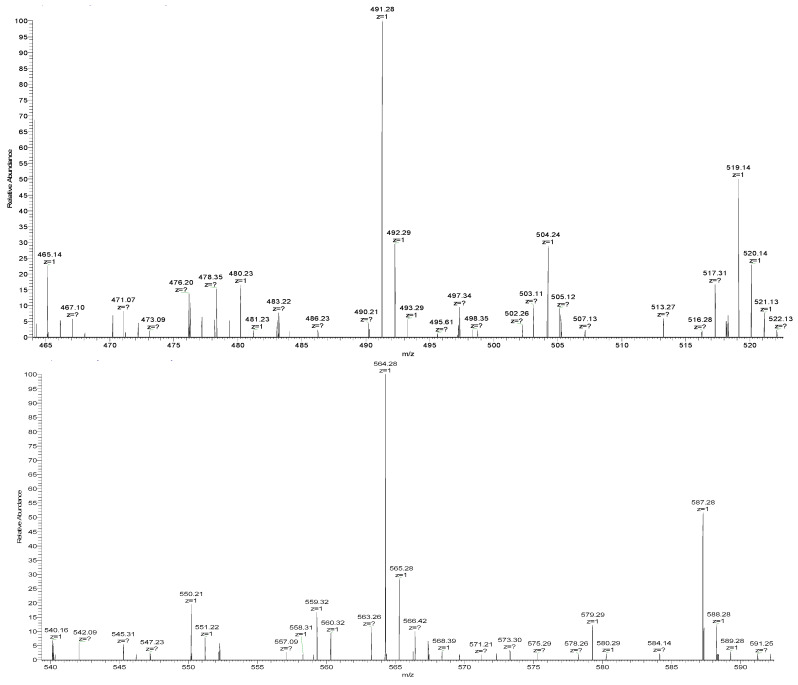
Secondary mass spectra of three novel ACE inhibitory peptides.

**Figure 14 foods-11-00394-f014:**
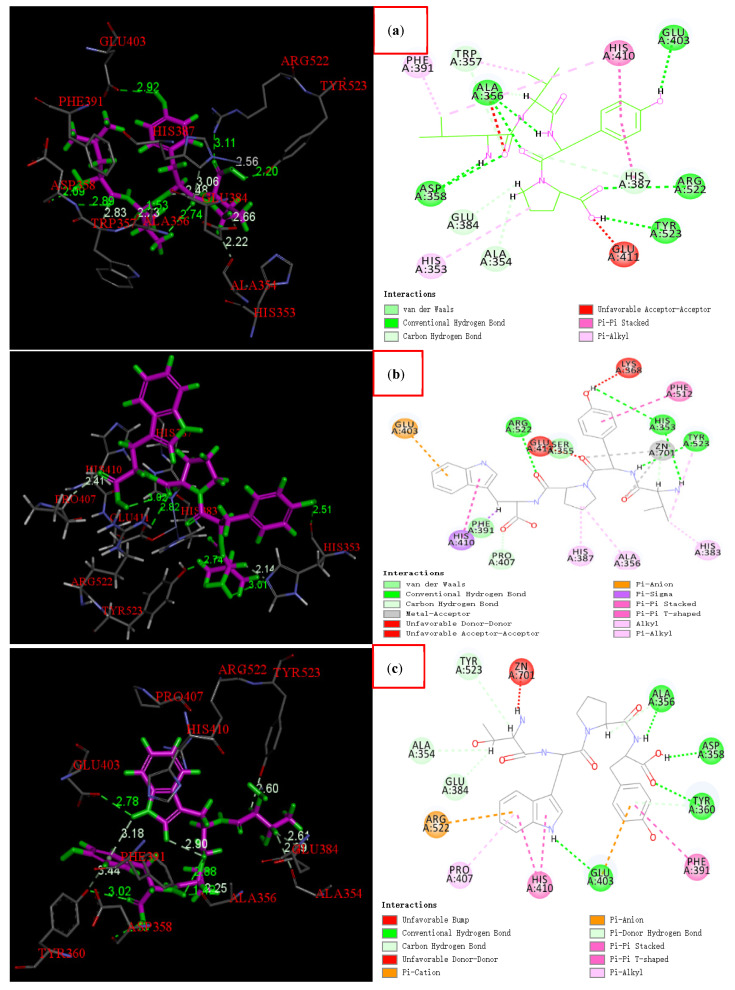
The docking results of three active peptide(LVYP(Leu-Val-Tyr-Pro), VYPW(Val-Tyr-Pro-Trp) and YPWT(Tyr-Pro-Trp-Thr)) molecules. The figure on the left represents the types of hydrogen bonds and their bond lengths, and the figure on the right represents all the linkages between the peptide and ACE, where (**a**) represents the combination mechanism diagram of LVYP and ACE; (**b**) represents the combination mechanism diagram of VYPW and ACE; (**c**) represents the combination mechanism diagram of YPWT and ACE.

**Figure 15 foods-11-00394-f015:**
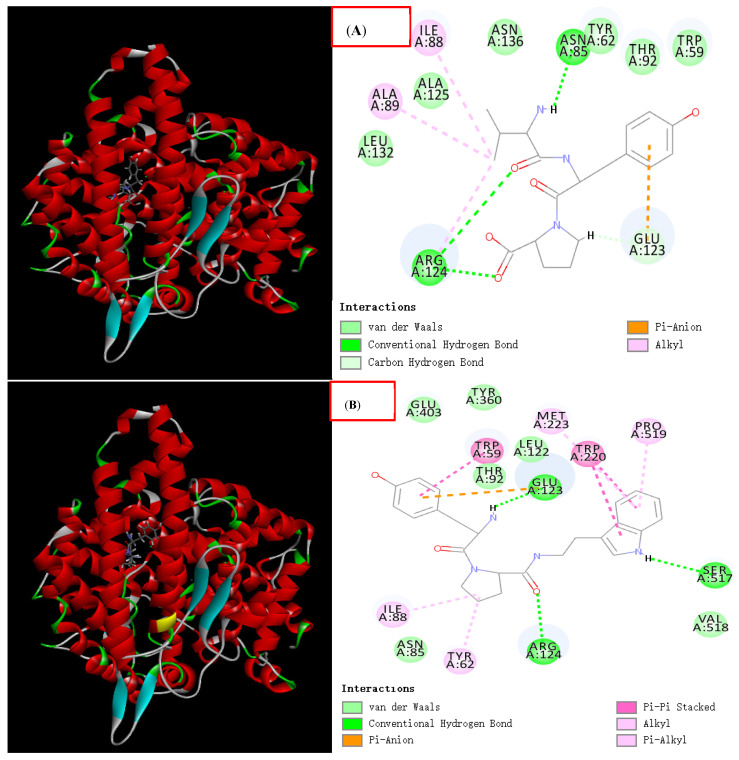
Molecular docking results for two shared tripeptides, VYP(Val-Tyr-Pro) and YPW(Tyr-Pro-Trp). (**A**) represents the binding mechanism between VYP and the receptor ACE. (**B**) represents the binding mechanism between YPW and the receptor ACE.

**Figure 16 foods-11-00394-f016:**
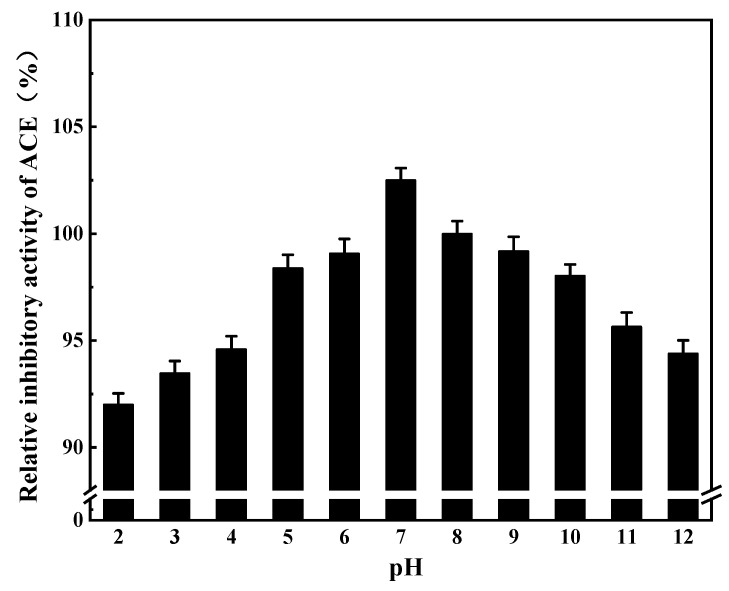
Stability of ACE inhibitory peptide at different pH levels.

**Figure 17 foods-11-00394-f017:**
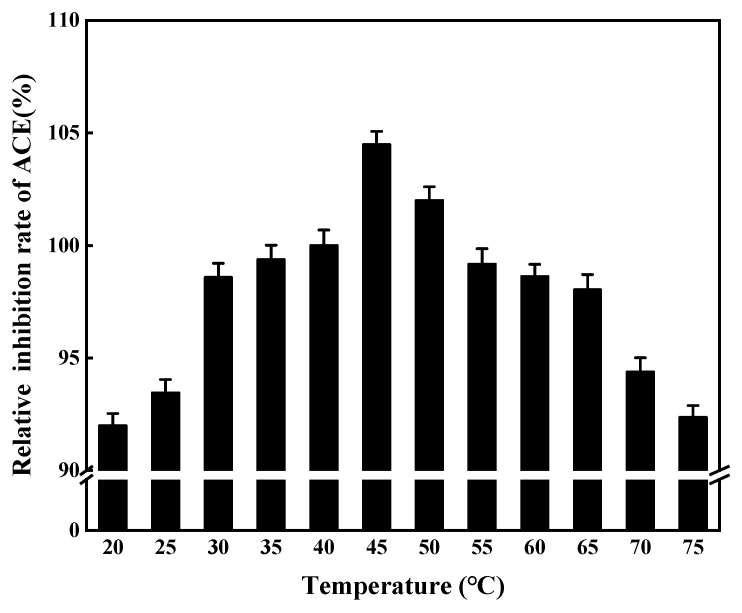
Stability of ACE inhibitory peptide at different temperatures.

**Figure 18 foods-11-00394-f018:**
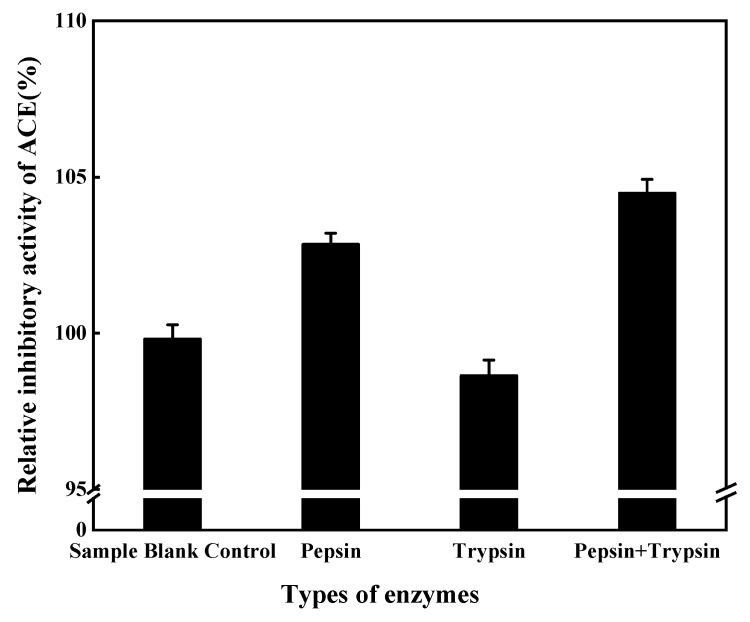
Effect of gastrointestinal digestive enzymes on ACE inhibitory peptides.

**Table 1 foods-11-00394-t001:** Coding values of response surface test factor levels.

Factor Level	Factor		
	Temperature of Enzymatic Digestion	pH	Time of Enzymatic Digestion
−1	40 °C	8.0	3.0
0	45 °C	8.5	3.5
+1	50 °C	9.0	4.0

**Table 2 foods-11-00394-t002:** Response surface test results.

Serial Number	pH	Temperature/°C	Time/h	Inhibition Rate of ACE/%
1	8.00	45.00	4.00	55.43
2	8.50	40.00	4.00	53.64
3	8.50	45.00	3.50	68.45
4	8.00	45.00	3.00	54.87
5	9.00	40.00	3.50	50.46
6	8.50	50.00	4.00	53.27
7	9.00	45.00	3.00	47.38
8	8.50	40.00	3.00	56.68
9	8.00	50.00	3.50	50.27
10	8.50	45.00	3.50	69.82
11	8.50	50.00	3.00	46.29
12	8.50	45.00	3.50	68.65
13	8.50	45.00	3.50	70.49
14	8.00	40.00	3.50	55.36
15	9.00	45.00	4.00	51.48
16	8.50	45.00	3.50	69.47
17	9.00	50.00	3.50	44.26

**Table 3 foods-11-00394-t003:** Analysis of variance.

Source	Sum of Squares	Degrees of Freedom	Mean Square	F Value	*p*-Value	Significance
Model	1288.34	9	143.15	337.78	<0.0001	***
A	62.44	1	62.44	147.34	<0.0001	***
B	60.78	1	60.78	143.41	<0.0001	***
C	9.24	1	9.24	21.81	0.0023	**
AB	0.31	1	0.31	0.73	0.4221	
AC	3.13	1	3.13	7.39	0.0298	*
BC	25.10	1	25.10	59.23	0.0001	***
A^2^	398.97	1	398.97	941.43	<0.0001	***
B^2^	384.35	1	384.35	906.94	<0.0001	***
C^2^	227.57	1	227.57	536.99	<0.0001	***
Residual	2.97	7	0.42			
Lack of Fit	0.14	3	0.045	0.064	0.9764	
Pure Error	2.83	4	0.71			
Cor Total	1291.31	16				

Significance: *p* < 0.05 means significant, indicated by “*”, *p* < 0.01 means more significant, indicated by “**”, *p* < 0.001 means extremely significant, indicated by “***”.

## Data Availability

The authors confirm that the data supporting the findings of this study are available within the article.

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
