# Peer review of "Optimisation and Characterisation of Novel Angiotensin-Converting Enzyme Inhibitory Peptides Prepared by Double Enzymatic Hydrolysis from Agaricus bisporus Scraps"

_foods, 2022, doi:10.3390/foods11030394_

Round 1
Reviewer 1 Report
The manuscript entitled “Optimization and characterization of novel angiotensin-converting enzyme inhibitory peptides prepared by double enzymatic hydrolysis from Agaricus bisporus scraps” by Rui Wang, Jianmin Yun, Shujuan Wu, Yang Bi, Fengyun Zhao is dealing with the topic of ACE(angiotensin-converting enzyme) inhibitory effect of four peptides derived from edible mushroom.
General comments
Angiotensin is one of the key compounds to control the human blood pressure. In this meaning, it is quite interesting to explore the compounds to regulate (inhibit) the angiotensin converting enzyme (ACE). The authors focused their attention on peptides derived from edible mushroom and find three peptides (LVYP, VYPW, and VPWT). The molecular docking studies suggested the formation of hydrogen binding with these peptides with S1 and S2 pockets of ACE.
- There are many studies dealing with the ACE inhibitory drugs (medicine) in the scientific field. The importance of this work is the effort to find out three peptides from edible mushroom by separating the fractions to inhibit ACE activity.
- The authors showed many figures to find out the suitable separating conditions, however, all these efforts are written in the supplementary section, not in the text.
- The authors did not discuss precisely the structure-activity relationship. It is highly desirable to discuss this point quoting previous works.
Specific comments
- The structure determination of three peptides is based on LC-MS/MS studies, however, it is quite important to measure other spectroscopic studies such as 1H-NMR, 13C-NMR and also two-dimensional NMR studies to clarify the precise structure-inhibitory activity.
- It is also worthy to prepare three peptides and examine their ACE activity.
- In LVYP and VYPW, VYP is the same sequence, however, the authors did not discuss anything on this point.
- In the case of YPWT, YPW sequence is the same with VYPW. It is highly desirable to discuss this point.
Author Response
Responds to the editor's and reviewer's comments:
- The authors showed many figures to find out the suitable separating conditions, however, all these efforts are written in the supplementary section, not in the text.
Response to comment: We thank the reviewer's suggestion. We have carefully checked the full text.
- The authors did not discuss precisely the structure-activity relationship. It is highly desirable to discuss this point quoting previous works.
Response to comment: We thank the reviewer's suggestion. We have supplemented the discussion of the structure-activity relationship quoting previous works, especially in combination with the latter two questions. “In LVYP and VYPW, VYP is the same sequence, however, the authors did not discuss anything on this point” and “In the case of YPWT, YPW sequence is the same with VYPW. It is highly desirable to discuss this point”. For details, please refer to the revised text (in discussion section), which have been marked in blue.
- The structure determination of three peptides is based on LC-MS/MS studies, however, it is quite important to measure other spectroscopic studies such as 1H-NMR, 13C-NMR and also two-dimensional NMR studies to clarify the precise structure-inhibitory activity.
Response to comment: We strongly agree with the reviewer's suggestion on this issue. Indeed, it is very important to measure other spectroscopic studies such as 1H-NMR, 13C-NMR and also two-dimensional NMR studies to clarify the precise structure-inhibitory activity of peptides. In fact, we have already started this work. However, since this paper focuses on reporting prepare novel peptides and examine their ACE activity. More in-depth studies on the structure and functional activity of these three peptides will be reported in subsequent manuscript.
Special thanks to you for your good comments again.

Reviewer 2 Report
Overall, the article is interesting and relatively well written. However, it requires work and clarification of certain things mainly in the materials and methods section. Detailed comments are included in the attached pdf file of the article.

Author Response
Responds to the editor's and reviewer's comments:
- It requires work and clarification of certain things mainly in the materials and methods section. Detailed comments are included in the attached pdf file of the article.
Response to comment: We thank the reviewer's suggestion. We have improved these things in the materials and methods section. For details, please refer to the revised text, which have been marked in red.
2.Explain why this temperature range has been chosen?
Response to comment: We thank the reviewer's suggestion. We preliminarily determined the general temperature range through pre-experiments, and these ranges meet the action conditions of the two enzymes. However, the specific temperature could not be determined. Therefore, the interval of 40, 45, 50, 55 and 60°C was determined by the averaging method for single factor experiment.
3.How the proteolysis was determined? The use of the TNBS method cannot be used as a measure of proteolysis but the amount of free amino acids. Proteolysis is a very complex process and the determination of free amino acids is only the determination of the compounds formed in its final stages.
Response to comment: We thank the reviewer's suggestion. We refer to the method in the article "Determination of the degree of hydrolysis of food protein hydrolysates by trinitrobenzene sulfonic acid. ". The content of free amino acids after enzymatic hydrolysis was used to determine the degree of protein hydrolysis.
4.It is difficult to establish a close relationship between the ACE-inhibitory activity in vitro and the hypotensive effect in vivo. This arouses some doubts concerning the use of the ACE-inhibitory activity in vitro as the sole criterion in the identification of substances with a potentially hypotensive effect, owing to the possibility of their physiological transformations in vivo. This has been confirmed in some study, where the authors demonstrated that the in vitro,ACE-inhibitory activity of naturally-formed bioactive peptides in foods caused no hypotensive effect in vivo.
Response to comment: Indeed, as the reviewer pointed out, it is difficult to establish a close relationship between the ACE-inhibitory activity in vitro and the hypotensive effect in vivo. Therefore, based on this consideration, we carried out the peptide’s stability verification test in this present work, and preliminarily investigated the effect of simulated human digestive tract environment (such as gastric acid, digestive enzymes and temperature) on the activity, and found that their relative activity was all greater than 90%.
- Please add missing spaces in the references.
Response to comment: We thank the reviewer's suggestion. We have added missing spaces in the references list.
Special thanks to you for your good comments again.
